# Biogenic Ferrihydrite Nanoparticles Produced by *Klebsiella oxytoca*: Characterization, Physicochemical Properties and Bovine Serum Albumin Interactions

**DOI:** 10.3390/nano12020249

**Published:** 2022-01-13

**Authors:** Nicoleta Cazacu, Claudia G. Chilom, Sorina Iftimie, Maria Bălășoiu, Valentina P. Ladygina, Sergey V. Stolyar, Oleg L. Orelovich, Yuriy S. Kovalev, Andrey V. Rogachev

**Affiliations:** 1Department of Electricity, Solid State and Biophysics, Faculty of Physics, University of Bucharest, RO-077125 Măgurele, Romania; s.nicoleta@yahoo.ro (N.C.); sorina.iftimie@fizica.unibuc.ro (S.I.); 2Department of Nuclear Physics, “Horia Hulubei” National Institute of Physics and Nuclear Engineering, RO-077125 Măgurele, Romania; balas@jinr.ru; 3Joint Institute for Nuclear Research, 141980 Dubna, Russia; orel@jinr.ru (O.L.O.); yukovalev@jinr.ru (Y.S.K.); rogachev@jinr.ru (A.V.R.); 4Moscow Institute of Physics and Technology, 141701 Dolgoprudniy, Russia; 5Federal Research Center KSC, Siberian Branch, Russian Academy of Sciences, 660036 Krasnoyarsk, Russia; ladvp@mail.ru (V.P.L.); stol@iph.krasn.ru (S.V.S.); 6Physics Department, School of Engineering Physics and Radio Electronics, Siberian Federal University, 660041 Krasnoyarsk, Russia; 7Kirensky Institute of Physics, Siberian Branch, Russian Academy of Sciences, 660036 Krasnoyarsk, Russia

**Keywords:** biogenic ferrihydrite nanoparticles, the binding mechanism, energy transfer, protein stability, molecular docking

## Abstract

The synthesis of nanoparticles inside microorganisms is an economical alternative to chemical and physical methods of nanoparticle synthesis. In this study, ferrihydrite nanoparticles synthesized by *Klebsiella oxytoca* bacterium in special conditions were characterized by scanning electron microscopy (SEM), energy-dispersive X-ray analysis (EDS), small-angle X-ray (SAXS), UV-Vis spectroscopy, fluorescence, fluorescence resonance energy transfer (FRET), and molecular docking. The morphology and the structure of the particles were characterized by means of SEM and SAXS. The elemental content was determined by means of the EDS method. The absorption properties of the ferrihydrite nanoparticles were investigated by UV-Vis spectroscopy. The binding mechanism of the biogenic ferrihydrite nanoparticles to Bovine Serum Albumin (BSA) protein, studied by fluorescence, showed a static and weak process, combined with FRET. Protein denaturation by temperature and urea in the presence of the ferrihydrite nanoparticles demonstrated their influence on the unfolding process. The AutoDock Vina and UCSF Chimera programs were used to predict the optimal binding site of the ferrihydrite to BSA and to find the location of the hydrophobic cavities in the sub-domain IIA of the BSA structure.

## 1. Introduction

The preparation of nanoparticles of certain sizes, shapes, and compositions is a complex task and an important area of research due to their various applications [1]. A promising perspective in this direction is the use of biological systems for the production of inorganic nanosized particles through green chemistry [2,3,4,5,6]. Microorganisms are well known for their ability to mineralize large amounts of iron under anaerobic conditions, accumulating as ferrihydrite in particular. Due to its high specific surface [7], ferrihydrite in the ultrafine state is a chemically active substance that interacts with a number of chemicals and organic molecules by the mechanism of surface adsorption/or co-precipitation.

As previously shown [8,9], ferrihydrite nanoparticles synthesized by the bacteria *K. oxytoca* during the biomineralization of solutions of iron salts from a nutrient medium exhibit unique magnetic properties: they are characterized by the antiferromagnetic order inherent in bulk ferrihydrite and spontaneous magnetic moments due to the spin decompensation in the sublattices of a nanoparticle. Thus, large applications in medicine and biotechnology [10,11] can be developed by the use of magnetic control of these nanoparticles [12]. Moreover, their properties can be improved by building functional structures with biomolecules, polymers or metals [13,14].

Biogenic ferrihydrite nanoparticles are synthesized by bacteria, such as *K. oxytoca* [8,15], *Ralstonia* sp. *SK03* [16], *Archaeoglobus fulgidus* [17], *Magnetospirillum gryphiswaldense* [18] or *Staphylococcus warneri* [19] under very strict and controlled conditions regarding size, shape, dimension and structure. These nanoparticles, usually coated with polysaccharides and protein traces, exhibit very low toxicity and good biocompatibility. In recent years, different physical and structural properties of biogenic ferrihydrite nanoparticles synthesized by the bacteria *K. oxytoca* were intensively investigated [15,20,21], due to the fact that bacterial *K. oxytoca* colonies reproduce easily in laboratory conditions and, therefore, can be used as a “biological factory” for the production of such nanoparticles [22,23].

The model protein for the transport and delivery of small molecules, albumin, is able to bind to various ligands, such as vitamins [24,25], flavonoids [26,27], drugs [28,29], and dyes [30]. This paper uses structural, spectroscopic, and computational methods in order to characterize the physicochemical properties of biogenic ferrihydrite nanoparticles produced by *Klebsiella oxytoca* and to investigate their capacity to bind serum proteins (albumin) for possible medical applications. Here, the structural properties and interactions of biogenic ferrihydrite nanoparticles with bovine serum albumin (BSA) are investigated. Examination of the absorption properties of the biogenic ferrihydrite nanoparticles in buffer solution, the binding mechanism to BSA, and protein stability in the presence of ferrihydrite nanoparticles were followed by spectroscopic and computational methods.

## 2. Materials and Methods

Synthesis of ferrihydrite nanoparticles. The bacterial colonies used in the present study were separated from the sapropel of Lake Borovoe (Krasnoyarsk Krai, Russia) by passing the taken sample through a magnetic separator. A bacterial biomass was grown under the microaerophilic and aerophilic conditions on a Lovley medium of the following composition: NaHCO_3_, 2.5 g/L; CaCl_2_ × H_2_O, 0.1 g/L; KCl, 0.1 g/L; NH_4_Cl, 1.5 g/L; and NaH_2_PO_4_ × H_2_O, 0.6 g/L. The ferric citrate concentration was equal to 0.5 g/L, and the yeast extract concentration was 0.05 g/L. The bacteria were cultivated at different illuminances, including complete darkness [15]. Usually, samples were collected after 7–90 days after the microorganisms had been inoculated into the culture medium. Some of the biomass obtained was used immediately after collection; another part was frozen for later use. To isolate magnetic particles, the bacterial biomass was centrifuged (10 min at 10,000 rpm) and then disrupted using a UZDN ultrasonic processor (1 min, 44 kHz, 20 W). The particles were recovered using a samarium-cobalt magnet and then dried at 40–80 °C [8].

According to the Mössbauer spectroscopic investigations [9], the biomineral nanoparticles produced by bacteria *K. oxytoca* contained two magnetically ordered phases. Each phase was characterized by two states of Fe^3+^ ions with close values of the quadrupole splitting. In the phase conventionally termed Fe12, the quadrupole splitting was equal to 0.6–1.0 nm/s. In the phase called Fe34, the quadrupole splitting lay in the range of 1.5–1.8 nm/s. According to the standard interpretation, the smaller values of the quadrupole splitting corresponded to weaker local distortions in the lattice. By varying the conditions for the cultivation of microorganisms (duration, illumination, medium composition), the nanoparticles for which the states of Fe^3+^ ions were identified from Mössbauer spectra either only as Fe12 or only Fe34 could be obtained [9,12]. The Fe12 and Fe34 powders studied in this work were separated from: (i) fresh biomass of bacterial colonies cultivated in a complete dark regime for 1 and 5 weeks, respectively, and (ii) frozen biomass of bacterial colonies cultivated in a complete dark regime for 1 and 3 weeks, respectively (Table 1).

Bovine serum albumin, BSA, (purity over 98%) was purchased from Merk company (Merk KGAA, Darmstadt Germany). BSA of the standard molar absorption coefficient at 280 nm (ε = 44,000 M^−1^ cm^−1^) was used for concentration measurements.

Buffer solution. All samples were prepared in 100 mM HEPES buffer (ROTH from Karlsruhe, Germany) (molecular weight of 238.3 g/mol), and the pH was set at 7.45 using an InoLab 720 pH-meter (WTW, Weilheim, Germany).

Scanning electron microscopy (SEM) images of the samples were obtained using a field-emission high-resolution scanning electron microscope, type FE-SEM SU-8020 (Hitachi, Tokyo, Japan) in function at FLNR-JINR, Dubna. Before the SEM analyses, the samples were covered with a thin layer of Au-Pd using the magnetron sputtering method (type Q150R ES).

Energy dispersive X-ray microanalysis (EDS) of the samples were accomplished with SEM device type S-3400N with EDS (Hitachi, Tokyo, Japan), FLNR-JINR, Dubna. Energy dispersive spectroscopy (EDS) is a micro-analytical technique conventionally used in scanning electron microscopy (SEM) for the local determination of chemical elements in solid samples [31]. A single EDS measurement means the acquisition of an energy-dispersive spectrum in which, (at discrete energies) the characteristic peaks of the chemical constituents are present. To determine the *i*th element concentration from the peaks obtained, it is necessary to process the spectrum [31]. The main outcome from the measurement is the value of the relative intensity of the spectral lines (*K*-ratio) measured after the optimization of the shape of the peaks. The concentration of the *i*th element in the specimen is calculated from Equation (1) [31]:(1)$$\frac{C_{i}}{C_{i}^{s}}=A\frac{I_{i}}{I_{i}^{s}}=A\cdot K-ratio$$
where: $\frac{C_{i}}{C_{i}^{s}}$ is the relative concentration in the specimen and in the standard; $\frac{I_{i}}{I_{i}^{s}}$ is the relative intensity of spectral lines; and A is the correction factor for the quantitative determination of the *i*th element.

Small-angle X-ray scattering (SAXS) measurements were performed on a Rigaku X-ray High Flux HomeLab instrument (Rigaku Corporation, London, UK) at MIPT, Dolgoprudny, Russia [32]. Using an X-ray wavelength of *λ* = 1.54 Ǻ and a transfer momentum, *Q*, the range of 0.005 ÷ 1.0 Ǻ^−1^ was obtained, where *Q* = (4π/*λ*)sin(*θ*/2) and *θ* is the scattering angle [33,34]. The data processing was accomplished with the SASView program [35]. The best overlap between the experimental curves was obtained based on the following relations (Equations (2)–(7)):(i)Guinier-Porod:
(2)$$I(Q)=\left\{\begin{array}{c} \frac{G}{Q^{s}}\exp\left[\frac{-Q^{2}R_{g}^{2}}{3-s}\right]\\ \frac{D}{Q^{m}} \end{array}\right\}\;\;\begin{array}{c} Q\leq Q_{1}\\ Q\geq Q_{1} \end{array}$$
where *G* and *D* are the Guinier and Porod scale factors, and $R_{g}$ is radius of gyration, m is Porod index, and *s* is a dimension variable, with values equal to 0 for spheres, 1 for rods and 2 for platelets.(ii)Ellipsoid model:
(3)$$I(Q)=\frac{scale}{V}F^{2}(Q,\alpha )+background$$
where $F(Q,\alpha )=\Delta \rho V\frac{3\left(\sin Qr-Qr\cdot \cos Qr)\right)}{{\left(Qr\right)}^{3}}$ for
(4)$$r={\left[R_{e}^{2}\sin^{2}\alpha +R_{p}^{2}\cos^{2}\alpha \right]}^{1/2}$$
in which α is the angle between the axis of the ellipsoid and the vector momentum transfer $\vec{Q}$; $V=\frac{4}{3}\pi R_{p}R_{e}^{2}$ is the volume of the ellipsoid; $R_{p}$ and $R_{e}$ are the polar radius along the rotational axis of the ellipsoid and equatorial radius perpendicular to the rotational axis of the ellipsoid, respectively; and Δρ is the scattering length density difference between the scatterer and the solvent.(iii)Triaxial ellipsoid: the scattering for randomly oriented particles of triaxial ellipsoidal form ($\frac{X^{2}}{R_{a}^{2}}+\frac{Y^{2}}{R_{b}^{2}}+\frac{Z^{2}}{R_{c}^{2}}$ is defined by the average over all orientations of the solid angle Ω by Equations (5) and (6):(iv)(5)$$I(Q)=scale{\left(\Delta \rho \right)}^{2}\frac{V}{4\pi }\int_{\Omega }\Phi^{2}(Qr)d\Omega +background$$
where $\Phi (Qr)=\frac{3\left(\sin Qr-Qr\cdot \cos Qr\right)}{{\left(Qr\right)}^{3}}$; $r^{2}=R_{a}^{2}e^{2}+R_{b}^{2}f^{2}+R_{c}^{2}g^{2}$;
(6)$$V=\frac{4}{3}\pi R_{a}R_{b}R_{c}$$


The *e*, *f,* and *g* terms are the projections of the orientation vector on the X, Y and Z-axes respectively.

The unified exponential-power law was calculated using Equation (7):
(7)$$I\left(Q\right)=\overset{N}{\underset{i=1}{\sum }}\left[G_{i}\;exp\left(-\frac{Q^{2}R_{gi}^{2}}{3}\right)+B_{i}\;exp\left(-\frac{Q^{2}R_{g\left(i+1\right)}^{2}}{3}\right){\left(\frac{1}{Q_{i}^{*}}\right)}^{P_{i}}\right]$$
where $Q_{i}^{*}=Q{\left[erf\left(\frac{QR_{gi}}{\sqrt{6}}\right)\right]}^{-3}$; $R_{gi}$ is the radius of gyration of the level; and $G_{i}$, $B_{i}$ and $P_{i}$ are the parameters to be chosen for each level.

UV-Vis absorption spectroscopy. A Perkin-Elmer spectrophotometer - Waltham, MA, US (located at the Faculty of Physics, Măgurele, Romania) was used to record the absorption spectra of the samples diluted in 100 mM HEPES buffer, at pH = 7.45. All investigations were operated at room temperature, in the 250–500 nm spectral range.

Steady state fluorimetry. A LS55 Perkin-Elmer fluorimeter - Waltham, MA, US (located at the Faculty of Physics, Măgurele, Romania) was used for the detection of the changes in the fluorescence emission of BSA, in the presence of biogenic ferrihydrite nanoparticles. The measurements were performed in 10 mm × 10 mm quartz cuvettes. The BSA concentration was kept constant at 3 µM. In thermal denaturation, the BSA was excited at 295 nm and the fluorescence emission was recorded in the spectral range of 310–500 nm. BSA denaturation (3 µM) by urea (0–5 M) in the presence of ferrihydrite nanoparticles (3 µM) was studied by excitation at 295 nm, and the fluorescent emission was recorded in the range of 310–500 nm.

Fluorescence resonance energy transfer (FRET). The efficiency of energy transfer (E) between the biogenic ferrihydrite nanoparticles and BSA was calculated with Equation (8) [36]:(8)$$E=1-\frac{F}{F_{0}}=\frac{R_{0}^{6}}{R_{0}^{6}+r^{6}}$$
in which *F*_0_ is the donor (protein) fluorescence intensity in the absence of the acceptor (ferrihydrite nanoparticles), *F* is the donor fluorescence intensity in the presence of the acceptor, *r* is the distance between the donor and the acceptor, and *R*_0_ is the Förster critical distance at which 50% of the excitation energy is transferred from the donor to the acceptor, which can be calculated from Equation (9) [36]:(9)$$R_{0}=9.78\;\times \;10^{3}\;{\left[(k^{2}n^{-4}Q_{D}J(\lambda ))\right]}^{1/6}$$
where *k* describes the relative orientation of the transition dipoles, *n* is the refractive index of the medium, *Q_D_* is the quantum yield of the donor fluorescence in the absence of the acceptor, and *J* is the overlap integral of the emission spectrum of the donor with the absorption spectrum of the acceptor, which can be determined according to Equation (10) [36]:(10)$$J=\frac{\int F_{D}(\lambda )\varepsilon_{A}(\lambda )\lambda^{4}d\lambda }{\int F_{D}(\lambda )d\lambda }$$
where *F_D_*(*λ*) is the corrected fluorescence intensity of the acceptor with the total intensity (area under the curve) normalized to unity, and *ɛ_A_* is the molar absorption coefficient of the acceptor at the wavelength λ.

Molecular docking. The crystalline structure of the apo BSA (PDB ID: 3V03) was retrieved from the RCSB Protein Data Bank [37,38,39]. Protein preparation procedures were performed as follows: adding hydrogen atoms, atom-type charges, ensured that missing side chains are added, molecule chain and missing bond breaks were detected and fixed, and bond orders were assigned.

The structure of the ferrihydrite was retrieved from the Crystallography Open Database [40,41]. The virtual screening software packages used for the docking of the ferrihydrite with BSA were PyRx [42] and UCSF Chimera, developed by the Resource for Biocomputing, Visualization, and Informatics from the University of California, San Francisco, with support from NIH P41-GM103311 [43]. The best energetic scoring function was generated with the Auto Dock Vina [44] algorithm. The apo form of BSA and the ferrihydrite crystal structures were opened in the PyRx virtual screening tool. During the simulation, the size of the grid box along the x, y and, z directions, was set to 25 Å^3^.

## 3. Results and Discussions

### 3.1. Morphological and Structural Characterization of the Biogenic Ferrihydrite Nanoparticles

#### 3.1.1. Scanning Electron Microscopy (SEM) Characterization

SEM images given in Figure 1 reveal several morphological similarities and distinctions among samples at different magnifications.

At small magnifications, the appearance of samples S2, S3, S4 is obviously different from that of sample S1. Sample S1 appears to be more compact, S2 has a branched twisted structure, S3 looks spongy, and S4 is scaly. At higher magnifications (i.e., ×100·10^−3^), different details of the samples are revealed. The compact agglomerates in sample S1 are composed of small formations of the order of tens of nanometers. Twisted, very thin layers form the S2 sample. On the surface of these layers, smaller structures than in the previous case were detected. In sample S3, the morphological aspect of this magnification reveals petal-like formations on the surface, on which particles of the order of nanometers are sprinkled. Comparing the investigated samples with particles produced in the same way by the metabolism of *Klebsiella oxytoca* bacteria cultivated but in a day–night illumination regime (Fe12, cultured 8 days; Fe34, cultured 21 days) [22], we found general similarities between the samples S1 and Fe12, and S3 and Fe34, respectively. Also, the distribution of such nanoparticles obtained in a water sol sample by dynamic light scattering (DLS) had a polymodal shape and modal values of 28.2, 105, 7, and 220.2 nm [45], which confirms the hierarchical structure obtained in Figure 1.

The morphological diversity of biogenic iron oxyhydroxide precipitates that have also been detected in Ordovician dolostone pores and subterranean cracks [46] in sediment samples from hydrothermal vents diffused along the seabed in the Tonga sector [47], in iceberg-hosted sediments [48], and in samples obtained from microorganisms such as *Ralstonia sp. SK03* [16], *Geobacter sulfurreducens* [49], *Shewanella putrefaciens* and *Geobacter metallireducens* [50], *Shewanella oneidensis MR-1* [51,52], etc., highlight the complex mechanisms of morphological development involved in the mineralization of Fe oxyhydroxide in nature. Their properties in many ways are closely related to their shape, size, and surroundings.

#### 3.1.2. Energy Dispersive Spectroscopy (EDS)

The energy dispersion spectra presented in Figure 2 allowed the detection of the existing compositional differences in the samples. For the analyzed samples, a total of 12 elements were determined.

In Table 2, the elemental contents depicted are as follows (concentration levels in descending order, c%): (i) O, Fe, C, P (c > 10%); (ii) Ca, K, N (S2 and S3), Cl (S2 and S4) (10% > c > 1%); (iii) Na, Mg, Si (S1 and S3), S (S2, S3 and S4), and Cl (S1, S3 and S4).

In the case of S1 (Fe12 from frozen biomass), several elements (N, S) were missing or their quantities were under the sensitivity threshold of the method. In the case of samples S1 and S2 (Fe12 from frozen biomass and F12 from fresh biomass), the major element revealed was O, and the elemental contents (in concentration levels in descending order, c%) were: O, Fe, C, P (tens of percent); Ca, N (only for S2), Cl (only for S2), and K (10% > c > 1%); and Na, Mg, Si (only for S1), S (only for S2) (c < 1%), and Cl (only for S1), respectively.

For samples S3 and S4 (Fe34 from frozen biomass and F34 from fresh biomass), the major element detected was O, with the elemental contents (in concentration levels in descending order, c%) being: O, Fe, C, P (tens of percent); Ca, Cl (only for S4), N (only for S3), K (10% > c > 1%); and Na, Cl (only for S3), Mg, S, Si (only for S3) (c < 1%), respectively.

Compared to the elemental content of another sample, S5 obtained from a fresh bacterial biomass grown in normal day-night lighting regime for 2 weeks and investigated earlier using X-ray fluorescence analysis (XRF) [53], in the case of cultivation in complete darkness, it can be seen that the oxygen concentration decreased significantly and carbon diminished. The exception was that in sample S2, Cl remained the same for S1 and S3, and increased for S2 and S4. New elements were detected, such as N, Na, Mg, Si, S (see Table 2). In addition, the content of Fe, P, Ca and K increased considerably (Table 2). Based on Mossbauer spectroscopy studies [53], it was established that the S5 sample had an intermediate structure compared to the specific structures for S1 and S3. Under these conditions, it becomes plausible to hypothesize that the lighting regime during cultivation has a major effect on the influence on the elemental composition. The following factors that seem to influence the composition are the age of the biomass (fresh or kept frozen) and the initial biomass duration of cultivation.

*Klebsiella oxytoca* strains, like other microorganisms, are known to produce exopolysaccharides [54], and some of them are specifically involved in the iron binding process [16,55]. Previous investigations of similar samples with S3 and S4 by FT-IR analysis [56] indicated the presence of glucose, polysaccharides, protein I and amine II [56,57]. This suggests that the iron-binding exopolysaccharides found in the analyzed samples are associated with protein residues and explains the presence of the diversity of elements found by EDX analysis. Earlier investigations of the nanoparticles obtained from the metabolism of *Klebsiell oxytoca* bacteria have shown the presence of poorly crystalline 2-line ferrihydrite and other oxyhydroxide compounds in different ratio [8,58].

#### 3.1.3. Biogenic Ferrihydrite SAXS Investigation

In Figure 3A–D the small-angle X-ray scattering experimental curves and their fits are presented. It can be seen that the SAXS structural features for all samples were different.

In the case of samples S1 and S3, two and three zones are detected on the experimental curves, respectively, which are best fitted with model curves. In the case of samples S2 and S4, the matching using the unified exponential/power law approximation [59] for one and two levels, respectively, satisfies the experimental data processing. Table 3 shows the determined dimensions of the obtained structures for all samples.

For sample S1, characteristic features detected by the SAXS method were that it contained nano-objects in the form of smooth-surfaced plates and ellipsoidal nano-objects of similar dimensions (~12 nm). In the case of sample S2, mass fractal structures and bigger clusters of ~29 nm ere determined. S3 had three types of structures, globular surface fractals (~29 nm), triaxial ellipsoids (~10 nm), and rod-shaped mass fractals of comparable size (14 nm). Sample S4 had the appearance of surface fractals containing large clusters (~33 nm) and small particles with a smooth surface (~1.5 nm).

### 3.2. Spectroscopic Approach to Investigate the Binding of Biogenic Ferrihydrite Nanoparticles to BSA

#### 3.2.1. UV-Vis Characterization of BSA-Biogenic Ferrihydrite Nanoparticles Complexes

The optical properties for ferrihydrite nanoparticles from S1–S4 samples diluted in 100 mM HEPES buffer at pH 7.45, and also for BSA, were investigated by UV-Vis spectroscopy (Figure 4). In the range of 200–400 nm, all the biogenic ferrihydrite nanoparticles strongly absorb UV radiation. This absorbance may be due to the following three factors: the presence of iron oxide nanoparticles (~340 nm), the presence of some traces of proteins containing Trp and/or Tyr (260–280 nm), and the presence of polysaccharides (~230 nm).

The conditions for obtaining biogenic ferrihydrite nanoparticles influenced their absorption properties. It was observed that ferrihydrite nanoparticles from frozen biomass incubated for 1 week (S1) showed a higher absorbance than those incubated for 3 weeks (S3), under the same conditions. This situation may be due to the fact that the incubation of bacteria for a longer period of time can lead to the removal of organic material (polysaccharides, proteins) on the surface of the nanoparticles. For the ferrihydrite nanoparticles from fresh biomass incubated for 5 weeks (S4), a higher absorbance than those incubated for 1 week (S2) was observed. The bacterial cultures from fresh biomass incubated in darkness for 1 week (S2) showed lower absorbance than those obtained from frozen biomass (S1).

#### 3.2.2. Characterization of the Fluorescence Quenching Mechanism of BSA by Ferrihydrite Nanoparticles

A fluorimetric approach was used to investigate the mechanism of the binding between biogenic ferrihydrite nanoparticles from the S1–S4 samples and BSA at room temperature. For this purpose, the Stern–Volmer Equation (11) was used:(11)$$F_{0}/F=1+K_{q}\tau_{0}[Q]=1+K_{SV}[Q]$$
where *Q* is the quencher, *F*_0_ and *F* are the fluorescence emission intensities of the protein (in the absence and in the presence of the quencher); *K_q_* is the bimolecular quenching rate constant; *τ*_0_ is the average lifetime of the protein without the quencher, and *K_SV_* is the Stern–Volmer constant (quenching constant).

From the Stern–Volmer plot (Figure 5A), the quenching constant, *K_sv_*, was determined. For the S1 sample, the *K_sv_* value was smaller than for S3 (Table 4). The bimolecular constant, *k_q_*, is the parameter that describes the type of quenching mechanism—static or dynamic. Considering the lifetime of BSA as 6.9 ns [60], we calculated the *K_q_* values for all four samples of biogenic ferrihydrite nanoparticles (Table 4). The quenching seemed to be initiated by the complex formation because the *K_q_* values were higher than 1 × 10^10^ M^−1^ s^−1^ (the limit of diffusion in aqueous solutions [36]).

To quantify the strength of ferrihydrite nanoparticles binding to BSA, the Scatchard Equation (12) was used.
(12)$$\log(\frac{F_{0}}{F}-1)=\log K_{b}+n\log[Q]$$
in which *K_b_* is the binding constant, [*Q*] is the ligand concentration, and *n* is the stoichiometry of the binding.

The Scatchard double logarithmic plot (Figure 5B) allows the determination of the *K_b_* values (Table 4). The biogenic ferrihydrite nanoparticles from all samples bound to BSA at one site (*n* = 1) and the binding was very weak.

#### 3.2.3. Fluorescence Resonance Energy Transfer between Bovine Serum Albumin and Ferrihydrite Nanoparticles (FRET)

Fluorescence resonance energy transfer (FRET) between the amphiphilic protein BSA (as donor) and ferrihydrite nanoparticles (as acceptor) was investigated. The FRET efficiency between ferrihydrite nanoparticles and the Trp residues from BSA is illustrated in Figure 6.

The efficiency of energy transfer, *E*, was determined with Equation (8), and the computed values for all samples are listed in Table 5. The overlap integral between the fluorescence emission spectrum of the donor (BSA) and the absorption spectrum of the acceptor (ferrihydrite nanoparticles), was calculated according to Equation (10), using the following parameters: *k*^2^ = 2/3, *n* = 1.336, and *Q*_D_ = 0.118. The *J* parameter is listed in Table 5, together with the value of *R*_0_, calculated with Equation (9). The distance between BSA and the biogenic ferrihydrite nanoparticles was in the range of 1.99–2.47 nm (0.5R < r < 1.5R). These results are in accord with the quenching results (Table 5), confirming the static mechanism of the binding between biogenic ferrihydrite nanoparticles and BSA.

#### 3.2.4. Stability of BSA in the Presence of the Ferrihydrite Nanoparticles

The stability of the BSA structure under the influence of temperature and urea was analyzed by fluorescence spectroscopy. These studies were necessary because ligand binding may influence the stability of protein conformations.

In the case of thermal denaturation, a decrease in fluorescent emission of BSA and BSA-biogenic ferrihydrite nanoparticles was observed with increasing temperature (Figure 7A). This is an indication of the unfolding of BSA conformation. For a more complex study of the denaturation temperature, Van ’t Hoff’s representation was used (Equation (13); Figure 7B).
(13)$$K_{eq}=\frac{\left[D\right]}{\left[N\right]}$$
where [*D*] and [*N*] are the denatured and the native states of BSA conformation. Applying the equilibrium condition *K_eq_* = 1, the denaturation temperature for BSA was determined. From the graphical representation of ln *K_eq_* vs. 1/T (Figure 7B), the unfolding temperature was obtained as the ratio between the slope and the interception with the ordinate. The results obtained are shown in Table 6 and suggest that the protein unfolds at a higher temperature in the presence of nanoparticles (the highest unfolding temperature value was obtained in the presence of S2 nanoparticles, with 6.65 °C greater than for BSA melting temperature, 49.26 °C). Another important aspect is the order in which the nanoparticles increase the stability of the BSA conformation: S2 > S1 > S4 > S3.

Regarding the chemical denaturation with urea, the structural stability of BSA and BSA- biogenic ferrihydrite nanoparticles was investigated by exciting BSA at 295 nm. With the increase of the urea concentration, a decrease of the BSA fluorescence emission was noticed which corresponds to a denatured state of the protein structure (Figure 8A). Nanoparticles from S1 do not seem to increase the stability of the protein while those from S2, S3, and S4 help the protein to denature at a higher concentration of urea (Figure 8B).

### 3.3. Exploration of BSA Binding Site by Docking

BSA ferrihydrite complexes were further analyzed by the AutoDock Vina program, to find the optimal binding site of ferrihydrite to protein. The optimization of the free energy determined from molecular mechanics force fields conducts to the best superimposition between ferrihydrite and the BSA site (Figure 9A). Following molecular docking, the best position of the ferrihydrite in the protein, with a binding affinity of Δ*G* = −4.4 kcal/mol (18.41 kJ/mol) and the apparent association constant of *K_a_* = 1.69 × 10^3^ M^−1^ were determined; this result was also confirmed earlier in fluorescence [20] and suggests a low affinity for the binding of ferrihydrite to BSA.

The ligand, ferrihydrite, is positioned in the BSA binding site, surrounded by Tyr 149, Arg 198, Trp 213, Arg 256, Gln 195, Glu 152, Lys 197, and Ser 191 (Figure 9B). The distance between Trp 213 and ferrihydrite is 0.89 nm, thus, the molecular docking results agree with these obtained in fluorescence and FRET. The binding of ferrihydrite occurs through hydrophobic cavities located in the sub-domain IIA of BSA (Figure 9A), according to other references [61].

## 4. Conclusions

This study aimed to characterize the structural properties and the binding mechanism of the biogenic ferrihydrite nanoparticles produced by *Klebsiella oxytoca* bacteria with bovine serum albumin, BSA.

Morphological and structural analyses performed by means of SEM and SAXS techniques showed specific differences at each hierarchical level between the samples. EDS revealed compositional differences that were more significant for the Fe, C, Cl, Ca, and K elements. Moreover, a small amount of N (S2, S3), Si (S1, S3), and S (S2–S4) appeared in some of the samples.

The conditions under which the bacteria were grown (darkness mode, culture time) and the biomass preparation when the particles were extracted (fresh, frozen) affected the dispersing properties of ferrihydrite nanoparticles.

The highest optical absorbance was observed for ferrihydrite nanoparticles from the frozen biomass of *Klebsiella oxytoca* bacteria incubated in a dark regime for 1 week (S1) and 5 weeks (S4). The quenching of BSA by biogenic ferrihydrite nanoparticles is a static process, but the complex formed is characterized by a weak constant. Of all the samples, S1 bound the most to the BSA site, with a moderate affinity. A FRET also occurred at a maximum distance of 2.47 nm. The molecular docking approach confirmed the spectroscopic results and showed that the binding site for the ferrihydrite was in hydrophobic cavities located in sub-domain IIA of the BSA. The BSA structure is stabilized in the presence of ferrihydrite nanoparticles, and the stability was influenced in the following order: S2 > S1 > S4 > S3. S1 does not stabilize the BSA structure against urea, but S2, S3, and S4 help BSA to denature at a higher concentration of urea. The results obtained are promising for future studies, for the understanding of the physical and chemical properties of ferrihydrite nanoparticles bound to albumin, and to explain the transport model protein.

The magnetic properties of ferrihydrite nanoparticles make them important, for example, in drug targeting applications and in magnetic hyperthermia. Another specific property of the ferrihydrite nanoparticles synthesized by *Klebsiella oxytoca* is the presence of the polysaccharide layer on their surface. It is known that polysaccharide complexes are used, for example, in the treatment of iron deficiency disease. Consequently, these properties lead to an increased interest in ferrihydrite particles in biomedical research.

## Figures and Tables

**Figure 1 nanomaterials-12-00249-f001:**
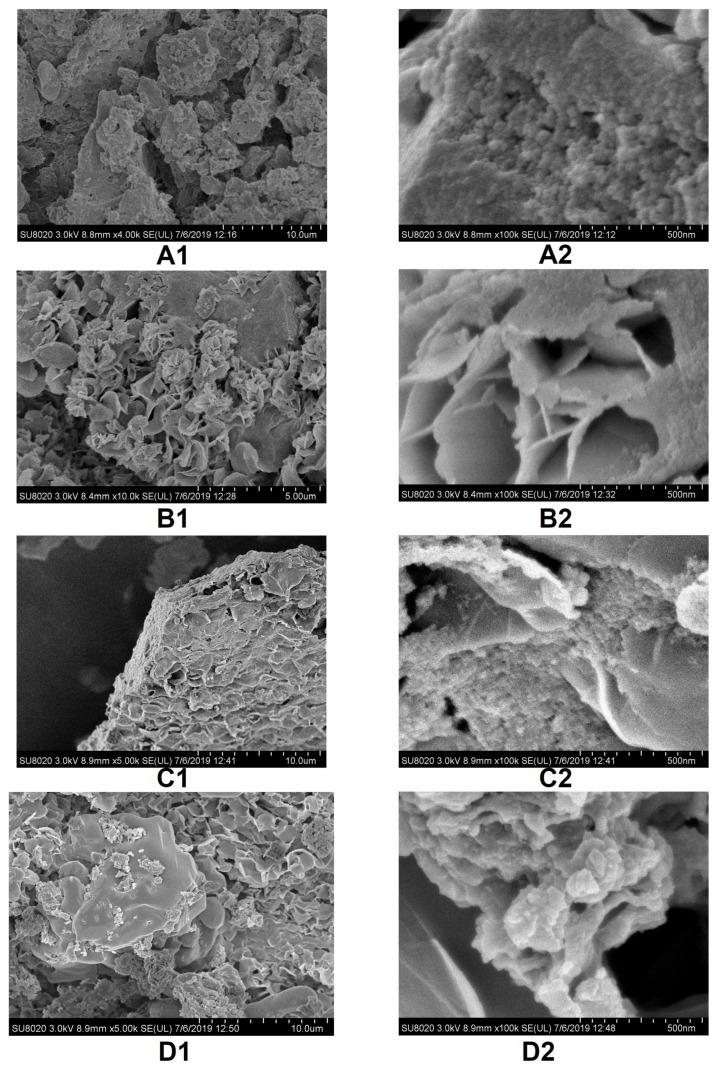
SEM micrographs of S1 (type F12, from frozen biomass, cultivated 1 week, ×4·10^−3^ (**A1**) and ×100·10^−3^ (**A2**) magnifications), S2 (type F12, from fresh biomass, cultivated 1 week, ×10·10^−3^ (**B1**) and 100·10^−3^ (**B2**) magnifications), S3 (type F34, from frozen biomass, cultivated 3 weeks, ×10·10^−3^ (**C1**) and ×100·10^−3^ (**C2**) magnifications), S4 (type F34, fresh biomass, cultivated 5 weeks, ×5·10^−3^ (**D1**) and ×100·10^−3^ (**D2**) magnifications) ferrihydrite particles.

**Figure 2 nanomaterials-12-00249-f002:**
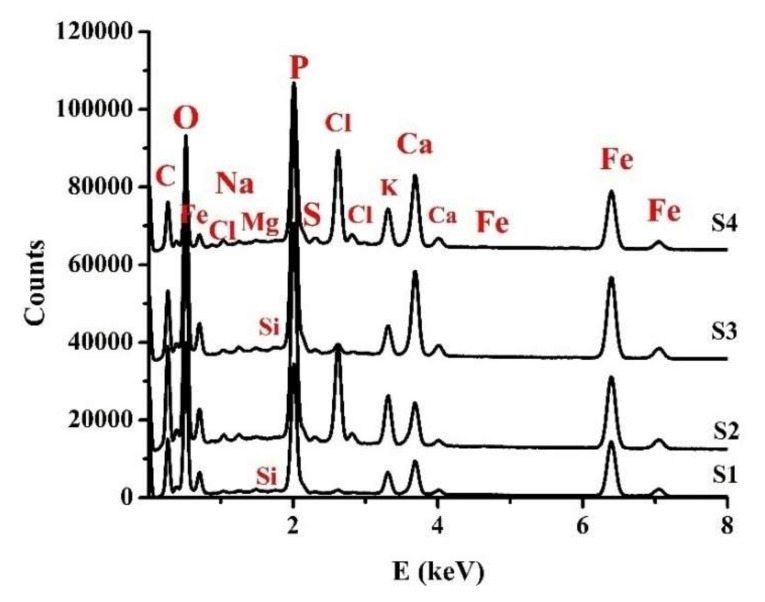
Energy-dispersive spectra of samples S1, S2, S3, and S4.

**Figure 3 nanomaterials-12-00249-f003:**
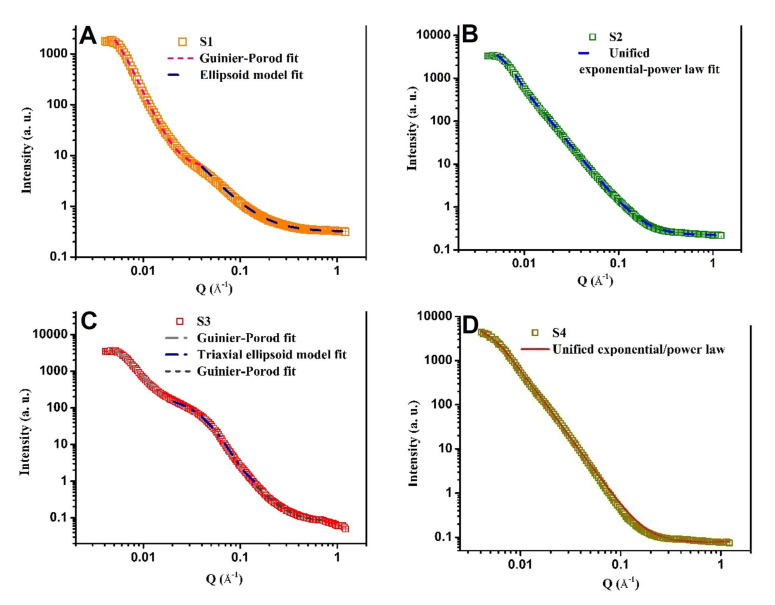
SAXS experimental and fitted curves of S1 (**A**), S2 (**B**), S3 (**C**), and S4 (**D**) samples.

**Figure 4 nanomaterials-12-00249-f004:**
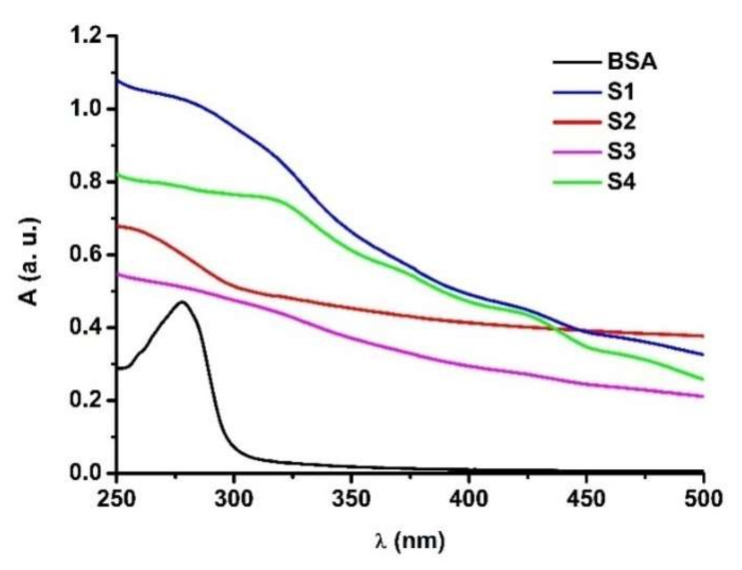
The absorbance of BSA (3 µM) and of S1–S4 (3 µM) samples at pH 7.45. Samples were studied in 100 mM HEPES buffer, at room temperature.

**Figure 5 nanomaterials-12-00249-f005:**
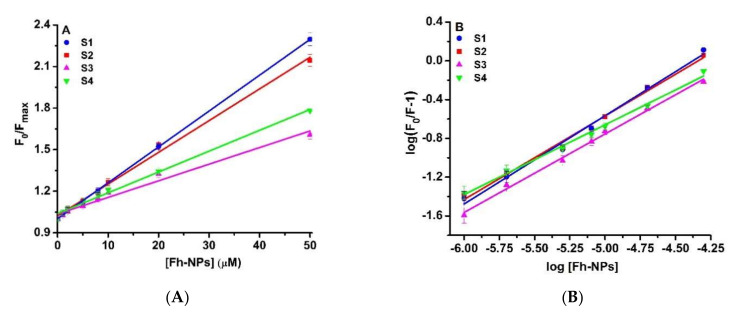
(**A**) The Stern-Volmer representation and (**B**) the Scatchard double logarithmic plot for the binding of biogenic ferrihydrite nanoparticles (0–50 µM) to BSA (3 µM) at room temperature.

**Figure 6 nanomaterials-12-00249-f006:**
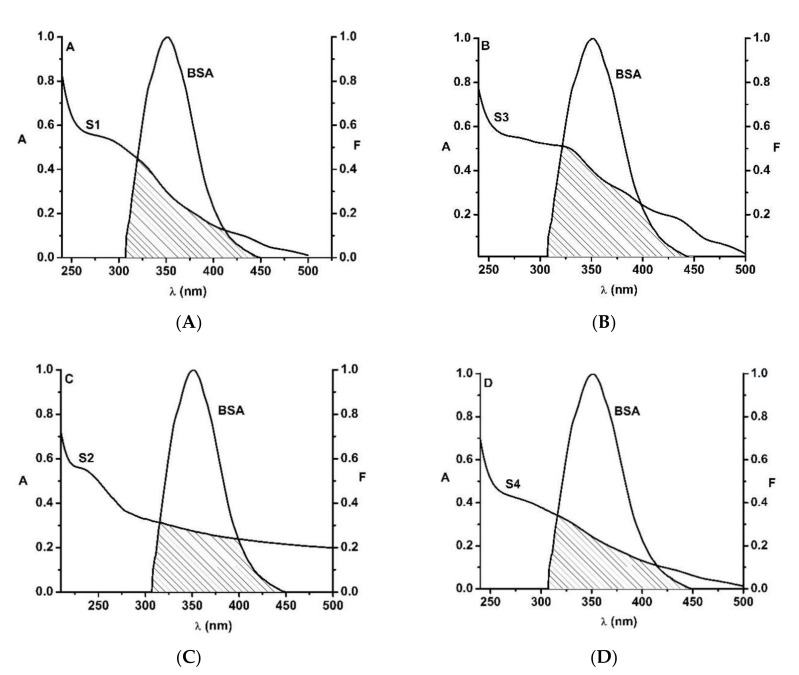
The overlap of the fluorescence emission spectrum of BSA and the absorption spectrum of biogenic ferrihydrite nanoparticles ([BSA]:[ferrihydrite nanoparticles] = 1:1) for the samples S1 (**A**), S2 (**B**), S3 (**C**) and S4 (**D**).

**Figure 7 nanomaterials-12-00249-f007:**
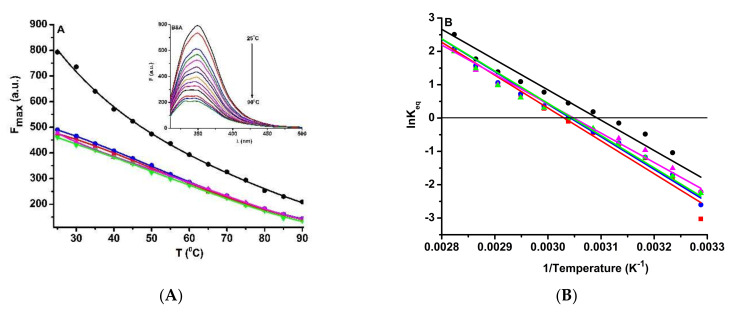
(**A**) BSA (3 µM) denaturation by temperature in the presence of the ferrihydrite nanoparticles (3 µM). (**B**) Graphical representation of ln*K_eq_* vs temperature, corresponding to the BSA (●), BSA-S1 (●), BSA-S2 (_▀_), BSA-S3 (▲), and BSA-S4 (▼) samples.

**Figure 8 nanomaterials-12-00249-f008:**
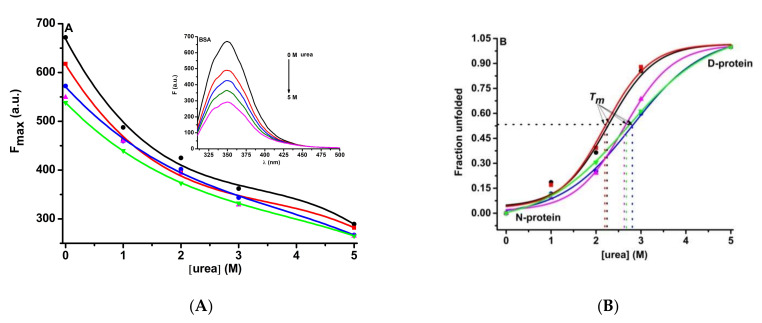
(**A**) BSA (3 µM) denaturation by urea in the presence of the ferrihydrite nanoparticles (3 µM). (**B**) Unfolding curves corresponding to: BSA (●), BSA-S1 (●), BSA-S2 (_▀_), BSA-S3 (▲), and BSA-S4 (▼) samples.

**Figure 9 nanomaterials-12-00249-f009:**
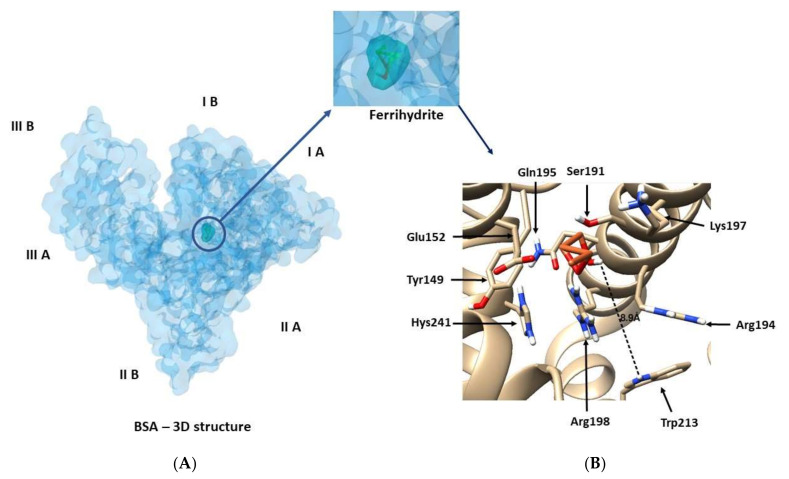
(**A**) The docking site of ferrihydrite in the surroundings of BSA residues. (**B**) The minimized average structure of the ferrihydrite (orange) docked into BSA molecules.

**Table 1 nanomaterials-12-00249-t001:** Experimental conditions for the synthesis of biogenic ferrihydrite nanoparticles.

Sample	Type	Cultivation Time	Illumination Regime	Type of Biomass
S1	Fe12	1 week	Dark	frozen
S2	Fe12	1 week	Dark	fresh
S3	Fe34	3 weeks	Dark	frozen
S4	Fe34	5 weeks	Dark	fresh

**Table 2 nanomaterials-12-00249-t002:** Results of the elemental analysis (in mass %) of biogenic ferrihydrite samples S1, S2, S3, S4, and S5 * [53].

Composition/Sample	S1	S2	S3	S4	S5 *
C	14.70	16.85	11.14	15.26	16.50
N	-	5.14	5.14	-	-
O	42.00	42.26	42.02	26.62	60.53
Na	0.43	0.54	0.36	0.75	-
Mg	0.14	0.25	0.26	0.14	-
Si	0.05	-	0.06	-	-
P	11.63	10.40	11.90	12.49	8.45
S	-	0.12	0.06	0.12	-
Cl	0.33	5.79	0.55	9.02	0.33
K	2.62	3.38	1.96	4.18	1.68
Ca	4.79	3.46	6.90	9.47	2.93
Fe	23.31	16.96	19.66	21.94	9.20
Other	-	-	-	-	0.38
Total	100	100	100	100	100

**Table 3 nanomaterials-12-00249-t003:** The morphology and dimensions of the detected nano-objects in the structures of samples S1–S4.

Sample	Q-Range	Fitting Model	Parameters (nm)
S1	0.004 ÷ 0.04	Guinier-Porod	Rg = 12.6 ± 0.2 S = 2 α = 4.03
	0.04 ÷ 1.2	Ellipsoid	R_a_ = 0.2 ± 0.05 R_b_ = 12.5 ± 0.2
S2	0.005 ÷ 1.2	Unified exponential-power law (1 level)	R_g_ = 29.0 ± 0.1 α = 2.65 ± 0.005
S3	0.005 ÷ 0.02	Guinier-Porod	Rg = 29.4 ± 0.1 S = 0 α = 3.52 ± 0.004
	0.02 ÷ 0.15	Triaxial ellipsoid	r_a_ = 5.5 ± 0.2 r_b_ = 10.5 ± 0.2 r_c_ = 2.7 ± 0.1
	0.15 ÷ 0.9	Guinier-Porod	Rg= 14.5 ± 0.1 S = 1 α = 2.95 ± 0.005
S4	0.005 ÷ 1.2	Unified exponential-power law (2 levels)	Rg = 12.6 ± 0.2 S = 2 α = 4.03

**Table 4 nanomaterials-12-00249-t004:** The binding parameters of the interaction of the biogenic ferrihydrite nanoparticles from S1–S4 samples and BSA.

Type of Nanoparticles	*KSV* (M^−1^)	*kq* (M^−1^ s^−1^)	*Kb* (M^−1^)	*n*
S1	0.90 × 10^6^	1.30 × 10^14^	13.36 × 10^3^	0.89
S2	0.81 × 10^6^	1.17 × 10^14^	6.37 × 10^3^	0.76
S3	4.16 × 10^6^	6.02 × 10^14^	1.34 × 10^3^	0.72
S4	3.55 × 10^6^	5.14 × 10^14^	0.87 × 10^3^	0.70

**Table 5 nanomaterials-12-00249-t005:** FRET parameters of the interaction of the biogenic ferrihydrite nanoparticles and BSA.

Type of Nanoparticles	*J* × 10^13^ (M^−1^ cm^−1^ nm^4^)	*E*	*R*_0_ (nm)	*r* (nm)
S1	2.31	0.19	1.91	2.42
S2	1.29	0.21	1.74	2.15
S3	1.23	0.29	1.72	1.99
S4	2.38	0.18	1.93	2.47

**Table 6 nanomaterials-12-00249-t006:** Thermodynamic fingerprint of BSA denaturation in the presence of biogenic ferrihydrite nanoparticles.

Sample	Δ*H* (kJ mol^−1^K^−1^)	ΔS (J mol^−1^K^−1^)	*T_m_* (°C)
BSA	75.582	233.70	49.26
BSA-S1	81.237	247.15	54.53
BSA-S2	81.960	248.38	55.82
BSA-S3	73.232	223.26	53.85
BSA-S4	80.478	245.12	54.16

## Data Availability

Not applicable.

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
