# Peer review of "Biogenic Ferrihydrite Nanoparticles Produced by Klebsiella oxytoca: Characterization, Physicochemical Properties and Bovine Serum Albumin Interactions"

_nanomaterials, 2022, doi:10.3390/nano12020249_

Round 1
Reviewer 1 Report
The manuscript by Cazacu et al. deals with the physico-chemical characterisation and interaction with bovine serum albumin (as model transport protein) of some biogenic ferrihydrite nanoparticles. Experimental characterization (SEM, EDS, SAXS) and spectroscopic (UV-Vis, fluorescence) methods are accompanied by molecular docking aiming to investigate the binding site of ferrihydrite.
I consider that the article can be published in nanomaterials after a minor revision is performed concerning the following points:
Page 5, line 185: The definitions given for J and FD(λ) are incorrect. J is the overlap integral of the emission spectrum of the donor and the absorption spectrum of the acceptor; FD(λ) is the corrected fluorescence intensity of the donor. The reference (probably Lakowicz) for the equations used to estimate FRET should be given.
Page 6: No discussion is made regarding the SEM micrographs. Is it possible to add a comment on the morphology of the ferrihydrite nanoparticles obtained in this study, compared to that of other biogenic ferrihydrite nanoparticles reported in the literature?
Page 10, line 308: The value of the excited state lifetime of BSA (τ0) used to compute kq should be 6.9 ns.
Page 11, Table 4: The notations used in the column headers (KSV x 106, etc.) imply that 0.90 = KSV x 106, resulting that KSV = 0.9 x 10-6 M-1, which is incorrect. The column headers should read KSV x 10-6, kq x 10-14, Kb x 10-3. See also J in Table 5.
Page 12, line 347: I suggest rephrasing the sentence “the binding of ligands contributes to the greater stability of a protein conformation.” Ligand binding to proteins can have either a stabilizing or a destabilizing effect.
The notation FhNPs is firstly introduced in page 12 and only used twice.
Reviewer 2 Report
The manuscript presents various calculation models for interpreting the results but contains also inconsistencies and only partly proven starting presumptions, thus major revision can be suggested.
The principal starting presumption is that the iron containig product produced by K. Oxytoca bacteria is ferrihydrite. In contrast, the elemental analysis presented in Table 2 shows rather large scatter by comparing the various samples. For instance, the smallest and largest O/Fe ratio are ca. 2 vs. 5, attesting that the samples are not homogeneous at all. Further, significant amounts of P and C are present, thus iron phosphates, carbonates or other organic complexes may also be present. In addition, iron may be present in reduced (Fe(II)) sate as well, since samples S1-S4 were kept in darkness (cf. ref. Kianpour et. al. 2018 who identifies the bacterial product as iron-exopolysaccharide-complex). Please provide more sound proves for exclusive presence of ferrihydrite or modify the text admitting the presence of other components as well.
In the abstract „dispersion of ferrihydrite nanoparticles was investigated by UV-Vis spectroscopy” is mentioned which might refer to distribution of size of nanoparticles in various samples. In contrast, the influence of the size distribution of nanoparticles is not considered all in the interpretation of results.
Further remarks:
- In Fig. 1 the morphologies of samples are compared. Please note in the legend, that the magnification for samples S3 and S4 is one order of magnitude larger than for S1 and S2.
- Please specify whether the values in Table 2 are in mass % or atomic %.
- AutoDock Vina program was also used to find the optimal binding site of ferrihydrite to BSA. However binding of only a single ferrihydridrite molecule is presented in Figure 9, in contrast to the whole previous part of the manuscript which is related to (presumed) ferrihydrite nanoparticles.
- In the reference listing please use either alphabetic order of the authors, or use serial numbers both in the text body and in the list.
Reviewer 3 Report
The article of Cazacu and coworkers describes the synthesis of ferrihydrite NPs by bacteria. The work is very interesting, the characterization is very extensive but this reviewer considers that will be very valuable to determine the size and the dispersion state of NP by DLS.
It will be also interesting to add some information about the applications that these NP will have.
Round 2
Reviewer 2 Report
The suggested improvements are considered and completed, the acceptance of the manuscript is proposed.
Reviewer 3 Report
Congratulations
This manuscript is a resubmission of an earlier submission. The following is a list of the peer review reports and author responses from that submission.